# Electrospun Nanofibers Based on Polymer Blends with Tunable High-Performance Properties for Innovative Fire-Resistant Materials

**DOI:** 10.3390/polym14245501

**Published:** 2022-12-15

**Authors:** Diana Serbezeanu, Corneliu Hamciuc, Tăchiță Vlad-Bubulac, Mihaela-Dorina Onofrei, Alexandra Bargan, Daniela Rusu, Dana Mihaela Suflet, Gabriela Lisa

**Affiliations:** 1“Petru Poni” Institute of Macromolecular Chemistry, Aleea Gr. Ghica Voda, 41A, 700487 Iasi, Romania; 2Department of Chemical Engineering, Faculty of Chemical Engineering and Environmental Protection, Gheorghe Asachi Technical University of Iasi, Bd. Mangeron 73, 700050 Iasi, Romania

**Keywords:** poly(ether-ether-ketone) (PEEK), DOPO-containing polyimide, thermal stability, mechanical properties, rheology

## Abstract

The main concern of materials designed for firefighting protective clothing applications is heat protection, which can be experienced from any uncomfortably hot objects or inner spaces, as well as direct contact with flame. While textile fibers are one of the most important components of clothing, there is a constant need for the development of innovative fire-retardant textile fibers with improved thermal characteristics. Lately, inherently fire-resistant fibers have become very popular to provide better protection for firefighters. In the current study, the electrospinning technique was applied as a versatile method to produce micro-/nano-scaled non-woven fibrous membranes based on various ratios of a poly(ether-ether-ketone) (PEEK) and a phosphorus-containing polyimide. Rheological measurements have been performed on solutions of certain ratios of these components in order to optimize the electrospinning process. FTIR spectroscopy and scanning electron microscopy were used to investigate the chemical structure and morphology of electrospun nanofiber membranes, while thermogravimetric analysis, heat transfer measurements and differential scanning calorimetry were used to determine their thermal properties. The water vapor sorption behavior and mechanical properties of the optimized electrospun nanofiber membranes were also evaluated.

## 1. Introduction

Firefighting protective clothing (FPC) is represented in most cases by a multilayer construct [1] made up of woven and/or nonwoven assemblies based on conventional fibers such as cotton, wool, viscose, nylon 6.6, etc. [2], as well as on high-performance fibers such as aramid (Nomex^®^, Kevlar^®^) [3,4], polybenzimidazole (PBI^®^), polybenzoxazole (Zylon^®^), arimide (P84^®^, Kermel^®^), etc. [5]. Each fiber has its own set of advantages and limitations [6]. One fiber membrane may display powerful heat protection but may offer insufficient comfort to wear, while another may be comfortable but costly. In terms of protection, one fiber membrane may have high tensile strength but low heat resistance, whereas another may have exceptional heat resistance but unfavorable tensile strength. In terms of comfort, one material may offer a smooth feeling when wearing and handling but lack moisture absorption, while another material may have good moisture absorption but may lack handling [6,7]. A judicious selection of fiber blend membranes for FPC is difficult and necessitates a thorough understanding and correlation of the required properties. Thus, as new technologies and novelty fibers emerge, favorable blends with various fiber alternatives to achieve the best balance of price, comfort and protection are constantly developing.

Aromatic polyimides are a class of thermostable heterocyclic polymers characterized by unique properties such as high thermal and thermo-oxidative stability, good resistance to radiation and chemicals, excellent mechanical and electrical characteristics. These polymers are suitable for many high-performance applications in electronics, electro-insulating materials, the aerospace industry and membrane preparation [8,9,10]. However, due to their rigid structure, they are often infusible and insoluble in organic solvents, which limits their applications in different fields of industry. Therefore, many research efforts have been made to improve the processability of polyimides. It was found that the introduction of bulky groups along the polyimide backbones can increase the solubility of the polymers. These groups enhance the free volume of the polymer and reduce the interchain interactions, thus improving the polyimide’s solubility in different organic solvents [11]. Aromatic polyimides show high values of char yield when they were heated to high temperatures, being thus suitable for their use as flame-resistant materials. Their flame-retardant characteristics can be improved by the introduction of different groups into their structure [12]. The presence of phosphorus atoms in the chemical structure of polyimides, even in low concentrations, may decrease the flammability. Moreover, phosphorus atoms do not confer toxicity to polymers, and very toxic compounds do not result during polyimide thermal degradation [13,14]. For example, by the incorporation of phosphazene units into polyimide macromolecular chains, a substantial improvement in their flame-resistant properties was achieved [15].

To date, some efforts have been made to obtain polyimides with improved flame retardancy and enhanced solubility in organic solvents via the introduction of bulky phosphaphenanthrene units into the macromolecular architecture of polyimides. Aromatic polyamides and polyimides containing bulky pendent groups derived from the monomer 9,10-dihydro-9-oxa-10-phosphaphenanthrene-10-oxide (DOPO) were prepared and reported by Liu et al. starting from a diamine having two DOPO groups and various dianhydrides [16,17]. The presence of these groups improved the solubility of the polymers and decreased the initial decomposition temperature. An improvement in the weight loss rate, thermal stability, thermo-oxidative stability and heat-insulating properties in the high-temperature region was reported. Chatterjee et al. reported a series of polyimides with pendent DOPO groups derived from the diamine monomer 1,1-bis [2′-trifluoromethyl-4′-(4″-aminophenyl)phenoxy]-1-(6-oxido-6H-dibenz<c,e><1,2>oxaphosphorin-6-yl)ethane and different aromatic dianhydrides [18]. The authors concluded that the presence of DOPO groups increased the free volume of the polymers and the chain rigidity, being thus suitable for the preparation of polymer membranes for gas separation. Other polyimides containing DOPO groups were prepared in our group recently, and their characteristics were evaluated [19,20]. The polyimides exhibited good solubility in organic solvents and can be processed into thin, flexible films. They exhibited fluorescence in film or in solution. A thermal decomposition mechanism was proposed based on pyrolysis–gas chromatography–spectroscopy analyses. The studies carried out using microscale combustion calorimetry analyses showed low values for the parameters total heat release (THR), heat release capacity (HRC), and the peak of heat release (PHRR), and high values for char yield, in the range of 39–49%, suggesting good flame retardancy.

Electrospinning is a useful and simple method for producing fibers with good homogeneity, having the size from a few micrometers to tens of nanometers. This procedure has many advantages, being inexpensive and allowing the possibility to control the characteristics of the nanofibers such as diameter, orientation and composition [21,22]. Nanofiber membranes prepared by the electrospinning method have attracted special interest because they exhibit an excellent combination of properties that include good thermal stability and a high specific surface area [23,24,25,26,27,28]. They are used in various applications as conductive fibers, protective clothing, membranes for filtration, separators for lithium-ion batteries, proton-exchange and anion-exchange membranes [29]. The electrospinning process for the preparation of various nanofibers was explored in different studies, and the relationship between the characteristics of the nanofibers and different parameters of the process, such as solution viscosity, charge density, surface tension, polymer molecular weight, dielectric constant, polymer concentration, applied voltage, feed rate and needle-to-collector distance, was investigated. To date, several attempts have been made to prepare polyimide-based nanofibers. Topuz et al. developed hydrophobic nanofibrous mats from different 4,4′-(hexafluoroisopropylidene)diphthalic anhydride (6FDA)-based polyimides that exhibited high efficiency for oil absorption [30]. Flexible multifunctional electromagnetic interference shielding films on the basis of electrospun silver nanowire-polyimide have been prepared recently [31]. The authors demonstrated that the reported materials showed excellent mechanical properties, low thermal conductivity and superior flame-retardant properties.

To the best of our knowledge, no nanofibers based on phosphorus-containing polyimides have been prepared and reported in the literature until the subject draw our attention in the recent years [32]. Thus, uniform submicron- or nano-sized fibers based on a polyimide resulted from the reaction of two monomers, bis(3-aminophenyl) methyl phosphine oxide and 4,4′-(4,4′-isopropylidenediphenoxy) bis(phthalic anhydride), were prepared by electrospinning using highly viscous polyimide solutions. Different parameters of the electrospinning process, such as polymer solution viscosity, the applied voltage, feed rate, were investigated in relation to the polyimide nanofiber morphology. The diameters of the electrospun fibers increased from 58 nm to 347 nm as the concentration of the polyimide solution was raised from 10 to 24 wt%. Due to the presence of phosphorus atoms in the polyimide structure, the nanofibers exhibit a high char yield of 63% at 700 °C, suggesting very good flame resistance properties.

Polyether ether ketones (PEEKs) represent another class of high-performance polymers that are applied in various industrial applications [33,34]. They are characterized by excellent mechanical properties, high thermal resistance and a low dielectric constant [35]. PEEKs have semi-crystalline characteristics that hinder their wide applications, and therefore amorphous PEEKs have been developed. For example, PEEKs based on phenolphthalein monomer led to polymers with amorphous structures. Nanofibers based on phenolphthalein PEEK have been successfully prepared via the electrospinning technique using solutions of the polymer in *N*,*N*-dimethylformamide [33].

The blending of different polymers is an important method that allows, through a suitable combination of components, obtaining materials with desired properties. The blending of a PEEK and a polyimide can be considered a useful procedure to combine the properties of both polymers and to obtain materials with high thermal and mechanical properties and chemical resistance that can be used for high-performance applications. Blends of PEEKs and polyetherimides were reported as being polymeric materials with very useful characteristics. It was found that a PEEK is miscible with a polyetherimide in any concentration, especially when both polymers are amorphous [36,37,38,39]. To broaden the variety of such composites based on PEEK-polyimide mixtures and to benefit from the advantages of the phosphorus atom when it is incorporated into a polyimide-type structure, in the present article, for the first time, a series of polymer blends resulting from different ratios of a phosphorus-containing polyimide and a phenolphthalein-based PEEK have been utilized to manufacture nanofiber membranes via the electrospinning technique. The expectations for obtaining nanofibers with a good combination of mechanical, thermal and flame resistance properties have been highlighted by performing composition-properties correlations and detailed discussion.

## 2. Materials and Methods

### 2.1. Materials

Phenolphthalein, 4,4′-difluorobenzophenone (purity 98.5%), *N*-methyl-2-pyrrolidone (NMP) (Reagent Plus (purity > 98.5%), water < 0.1%), potassium carbonate (ACS reagent, 99%) and 4,4′-oxydiphthalic anhydride (purity 97%), were purchased from Sigma-Aldrich Chemie GmbH and used as received. The 9,10-Dihydro-9-oxa-10-phosphaphenanthrene-10-oxide (DOPO) (purity 97%), received from TCI (Japan), prior to use in the synthesis of the diamine (2-DOPO-NH_2_), was dehydrated under vacuum for 5 h at high temperature (120 °C). The diamine (2DOPO-NH_2_) was synthesized according to published procedures starting from 4,4′-diaminobenzophenone and DOPO [16,19]. The other solvents used in the synthesis were of analytical grade and used as received from national or international companies.

### 2.2. Synthesis of PEEK

The synthesis of PEEK (Figure 1a) was conducted in a three-neck flask equipped with a magnetic stirrer, Dean-Stark and condenser, under nitrogen atmosphere. The three-neck flask was charged with phenolphthalein (3.498 g, 0.011 mol), 4,4′-difluorobenzophenone (2.398 g, 0.011 mol), potassium carbonate (0.022 mol), toluene (10 mL) and NMP (27 mL). The reaction mixture was heated under stirring to 140–150 °C and water was removed by azeotropic distillation with toluene for 5 h. Toluene was then removed from the reaction flask by distillation and the resulting mixture was heated at 180 °C for 8 h. After cooling to room temperature, the viscous polymerization mixture was diluted with NMP (30 mL) and the PEEK was precipitated by pouring into a diluted solution of HCl. The precipitated PEEK was filtered off, washed several times with hot water and vacuum dried at high temperature (100 °C) for 8 h. GPC: Number average molecular weight (M_n_) = 51,000 g/mol, PDI = 1.89. ^1^H NMR (DMSO-d_6_, ppm): 7.92 (2H, d), 7.84 (1H, s), 7.72 (5H, s), 7.39–7.38 (4H, d), 7.13–7.08 (8H, s).

### 2.3. Synthesis of Phosphorus-Containing Polyimide (PI-1)

The polyimide PI-1 presented in Figure 1, was prepared by two steps polycondensation reaction in solution of an equimolecular amount of diamine 2DOPO-NH_2_ and 4,4′-oxydiphthalic dianhydride as previously presented [19,20]. FTIR: 1780 and 1738 cm^−1^ (C=O asymmetrical and symmetrical stretching vibrations), 3444 cm^−1^ (–OH stretching vibrations), 1264 and 1063 cm^−1^ (C–O–C), 1463, 1117 and 792 cm^−1^ (P–Ph, P=O and P–O–Ph). GPC: Number average molecular weight (M_n_) = 13,000 g/mol, PDI = 1.55. ^31^P NMR (DMSO-*d6*, ppm): 29.90, 27.82

### 2.4. Preparation of PEEK/PI-1 Solution for Electrospinning

The PEEK/PI-1 solutions to be used in the electrospinning equipment were prepared by dissolving PEEK film in NMP at room temperature for 24 h under vigorous stirring. The obtained PEEK stock solution (15 wt%), which was the reference sample, denoted as PEEK/PI-1 (0), was used further to prepare PEEK/polyimide blends by adding different amounts of PI-1. Portions of 2 mL were taken from the PEEK stock solution and 0.05 g, 0.10 g, 0.15 g, 0.20 g and 0.25 g of phosphorus-containing polyimide PI-1 were introduced in each portion, to give PEEK/polyimide blended solutions denoted as PEEK/PI-1 (0.05), PEEK/PI-1 (0.1), PEEK/PI-1 (0.15), PEEK/PI-1 (0.20) and PEEK/PI-1 (0.25), respectively. The as-prepared solutions were utilized for performing rheology studies and then, they were subjected to electrospinning procedure to obtain nanofibrous membranes. 

### 2.5. Electrospinning Process

The electrospinning process of the homogeneous PEEK/PI-1 solutions in NMP was carried out using a Fluidnatek^®^ LE-50 laboratory line from Bioinicia S.L. (Valencia, Spain). The working parameters were as follows: voltage of ±22 kV, the distance between the tip of the needle and the collector was approx. 20 cm and the flow rate of the solution was approx. 20 µL∙min^−1^. The electrospun PEEK/PI-1 fibers were collected on a backer foil sheet attached to a static copper grid used as collector. The time for collecting the fibers was about 2 h. The working temperature was 25 °C.

### 2.6. Measurements

#### 2.6.1. Rheological Study

Rheological measurements on PEEK/PI-1 (0), PEEK/PI-1 (0.05), PEEK/PI-1 (0.1), PEEK/PI-1 (0.15), PEEK/PI-1 (0.20) and PEEK/PI-1 (0.25) solutions were performed using a CS-50 Bohlin rheometer with cone plate geometry (Malvern Instruments, cone angle of 4° and a diameter of 40 mm). The tests were performed at a temperature of 25 °C, over the 0.1–100 s^−1^ shear rate domain. The strain sweep tests were performed at a frequency range of 1 Hz over a strain range of 0.5–10 Pa. As a result, a shear stress of 2 Pa was chosen, and the oscillatory shear measurements were conducted between 0.1 and 100 Hz. For various measurements, rheological tests were obtained with a 5% accuracy.

#### 2.6.2. FTIR Spectroscopy

A LUMOS Microscope Fourier Transform Infrared (FTIR) spectrophotometer (Bruker Optik GmbH, Ettlingen, Germany), equipped with an attenuated total reflection (ATR) device was used to record the scans between 4000 and 500 cm^−1^ at a resolution of 4 cm^−1^ for the PEEK powder, PI-1 powder and the electrospun PEEK/PI-1 fibers.

#### 2.6.3. NMR Spectroscopy

^1^H NMR and/or ^31^P NMR was applied to confirm the chemical structure of the studied monomers and polymers. The tests were performed on a Bruker Avance DRX400 spectrometer, at various operating frequencies (400 MHz for ^1^H NMR, respectively 62MHz for ^31^P NMR), in DMSO-*d_6_*. The measurements were taken at room temperature.

#### 2.6.4. Scanning Electron Microscopy

Scanning electron microscopy (SEM) investigations for the obtained electrospun fibers were performed on a Verios G4 UC Scanning Electron Microscope (Thermo Scientific, SEM, FEI Company, Brno, Czech Republic). The electrospun PEEK/PI-1 fibers were coated prior examination with 6 nm platinum using a Leica EM ACE200 Sputter coater in order to increase the signal-to-noise ratio throughout SEM imaging and the electrical conductivity of the samples, and also to reduce the charging effects which appear during exposure to the electron beam. Therefore, high-resolution images can be achieved using this tool. SEM analyses were conducted in High Vacuum mode using a secondary electron detector (Everhart-Thornley detector, ETD) with 10 kV accelerating voltage. The diameters of the electrospun fibers were measured by means of the Image J program. At least 25 electrospun PEEK/PI-1 fibers from each sample, were taken into consideration to obtain the average diameters.

#### 2.6.5. BET Analysis

Dynamic moisture sorption capacity for the electrospun PEEK/PI-1 fibers has been determined using the fully automated gravimetric device IGAsorp made by Hiden Analytical, Warrington (UK). Before the sorption measurements started, the samples were dried at 25 °C, in flowing nitrogen (250 mL∙min^−1^) until the weight of the sample was in equilibrium at RH < 1%. Then, the relative humidity (RH) was progressively increased from 0 to 90%, in 10% humidity steps, every step having a pre-established equilibrium time between 40 and 60 min and the sorption equilibrium was obtained for each step. After that, the RH decreased and the desorption curves were registered. 

#### 2.6.6. Differential Scanning Calorimetry

Differential scanning calorimetry (DSC) measurements for the electrospun PEEK/PI-1 fibers were carried out on a Mettler Toledo DSC-type device (Mettler Toledo, Greifensee, Switzerland), by heating 3–5 mg of each sample, from 25 to 250 °C, with a heating rate of 10 °C∙min^−1^, under nitrogen atmosphere. The second heating cycle was used to determine the glass transition temperature (*T_g_*) values of the electrospun PEEK/PI-1 (0–0.25) fibers. *T_g_* was measured as the mid-point temperature of the change in the slope of the DSC signal.

#### 2.6.7. Thermogravimetric Analysis

Thermogravimetric analysis (TGA) of the electrospun PEEK/PI-1 fibers was investigated by using a Mettler Toledo TGA-SDTA851^e^ equipment (Mettler Toledo, Greifensee, Switzerland), in nitrogen atmosphere and a heating rate of 10 °C∙min^−1^, in the temperature range of 25–900 °C. The mass of the tested samples was in the range of 1.8–2.4 mg. 

#### 2.6.8. Mechanical Testing

The mechanical testing was performed using a Texture Analyser (Brookfield Texture PRO CT3^®^, Brookfield Engineering Laboratories Inc., Middleborough, MA, USA) at room temperature, following the ASTM D882 standard. 

#### 2.6.9. Heat Treatment

The heat treatment of the electrospun PEEK/PI-1 fibers was made in an oven preheated to 260 °C. The electrospun PEEK/PI-1 fibers were placed in the oven and kept at this temperature for 20 min. After 20 min, the electrospun PEEK/PI-1 fibers were taken out of the oven and subjected to SEM investigations.

#### 2.6.10. Heat Transmission

Evaluation of heat transmission through the electrospun PEEK/PI-1 fiber membranes has been conducted on a Protective Clothing Heat Transmission Index Tester (Yuyang Industrial Co., Model YY160, Dongguan, China). The equipment is designed, developed and manufactured according to ISO 9151 and EN 365 [40,41]. 

## 3. Results and Discussion

### 3.1. Dynamic and Steady State Rheology

Generally, the rheological behavior of a polymeric system presents a significant contribution to establishing the properties of the final products with large usage in various areas.

In this context, the flow behavior of PEEK/PI-1 (0) and PEEK/PI-1 (0.05; 0.10; 0.15; 0.20; 0.25) solutions in NMP, where a complex behavior appears under specific conditions of blend composition, was observed from the dynamic viscosity versus shear rate dependence (Figure 2). As can be observed, the Newtonian regime appears for PEEK/PI-1 (0) and for all PEEK/PI-1 (0.05; 0.10; 0.15; 0.20; 0.25) blend compositions for the whole studied shear rate domain. These results indicate significant changes in the PEEK/PI-1 systems as a result of the specific molecular rearrangement of the polymeric chains in blends. 

Therefore, the flow behavior illustrated in Figure 2 is depicted in the values obtained for the flow behavior (*n*) and consistency (*K*) indices (Table 1), which are approximated from a variation of shear stress (σ) with shear rate ($\dot{\gamma }$) according to the Ostwald–De Waele model (Equation (1)).
(1)$$\sigma =K\cdot {\dot{\gamma }}^{n}$$

Withal, literature estimates a Newtonian behavior of fluids for n = 1, a thinning behavior for n < 1 and a thickening behavior for n > 1 [42,43]. 

According to Figure 3, the curve shape, namely, the double logarithmic plot of shear stress versus shear rate, shows a similar behavior. As seen in Table 1, the rheological model corresponds to a Newtonian behavior (“n” close to 1), and the values of the consistency index increase for all blend compositions of the PEEK/PI-1 studied system. Consequently, these results can be explained by a modification of the specific interactions generated by the addition of PI-1 in the system.

The above-mentioned finding about the influence of structural properties of PEEK, PI-1 and variations in PI-1 composition on the mobility of the chains in the shear field was also evidenced by the investigation of the storage, G′, and loss, G″, moduli (Figure 4). In addition, the domain in which the system behaves as a viscous material is larger than the elastic domain, G″ > G′ [44,45,46]. Thus, one can remark that the overlap frequencies display lower values for PEEK in NMP and become higher when the PI-1 content in the polymer blends increases. Above this composition, the system exhibits a viscous flow regime greater than the elastic one, an aspect highlighted by the overlap frequencies, which move to higher frequency values, as shown in Figure 4 (e.g., $f_{G^{\prime }=G^{\prime \prime }}=22.37$ Hz for PEEK/PI-1(0.15) compared to $f_{G^{\prime }=G^{\prime \prime }}=43.94$ Hz for PEEK/PI-1 (0.25)). Moreover, the obtained results regarding the viscoelastic parameters indicate that the content of 0.15 g PI-1 represents the optimal weight fraction at which PI-1 behaves as a plasticizer for the PEEK.

Therefore, this behavior results from the chemical structure and composition of the polymers in the blend. It can be concluded that the rheological profiles obtained as a function of PI-1 content can help to examine the plasticization efficiency and the final properties of the studied systems.

### 3.2. Structural Identification

The chemical structures of the PEEK, PI-1 and the electrospun PEEK/PI-1 fibers were investigated by FTIR, ^1^H NMR and ^31^P NMR spectroscopy. Figure 5 exhibits the FTIR spectra of electrospun PEEK fibers and electrospun PEEK/PI-1 fibers. The absorption bands characteristic for electrospun PEEK were present in the FTIR spectrum of the neat sample and in all the composite mixtures. These bands are located at 1767 cm^−1^ and 1646 cm^−1^ (C=O stretching vibration), 1593 cm^−1^ (skeletal in-plane vibration of phenyl ring), 1495 cm^−1^ (aromatic rotations vibration), 1235–1165 cm^−1^ (diphenyl ether group, C–O–C rotation and stretch vibration), 928 cm^−1^ (aromatic out-of-plane bending vibration), 842 and 754 cm^−1^ (C–H out-of-plane bending substitution patterns). By the introduction of P-containing polyimide, characteristic absorption bands appeared in the FTIR spectra of the composites. Thus, the absorption bands at 1720, 1375 cm^−1^ (marked area in Figure 5) and 928 cm^−1^ (overlapped with 928 cm^−1^ (aromatic out-of-plane bending vibration) of PEEK) indicated the presence of phosphorus-containing polyimide PI-1 in the composite electrospun fibers. The absorption peaks around 1470 cm^−1^ (P-Ph), 1239 cm^−1^ (–P=O) and 1161 and 925 cm^−1^ (P–O–Ph), characteristic of the DOPO groups, were observed in the electrospun PEEK/PI-1 fibers and proved the presence of the DOPO group being incorporated into the electrospun PEEK/PI-1 fibers.

### 3.3. Electrospun PEEK/PI-1 Fibers Morphology

In order to investigate the structural morphology of the electrospun PEEK/PI-1 fibers, the SEM measurements were used. Moreover, SEM images were used for the measurements of the average fibers’ diameters by using the ImageJ program. In Table 2, the electrospinning condition and the average fiber diameters for the electrospun PEEK/PI-1 fibers are presented. 

The average fibers’ diameters of the electrospun PEEK/PI-1 fibers were between 211 ± 48 nm and 304 ± 45 nm. The microscope images of the electrospun PEEK/PI-1 fibers are shown in Figure 6. It can be observed that the morphology of the electrospun PEEK/PI-1 fibers was smooth and beaded free, with the average fibers’ diameters increasing as the content of PI-1 increased. The PEEK, 15 wt % solution in NMP, leads at electrospun PEEK/PI-1 (0) fibers with the average fibers’ diameters of 211 ± 48 nm. Meanwhile, the addition of the PI-1 in the PEEK solution leads to an average fibers’ diameters equal with 304 ± 45 nm (PEEK/PI-1 (0.25). As observed from Table 2 the average fibers’ diameters of the electrospun PEEK/PI-1 fibers increased as the fraction of PI-1 in the precursor solution increased. Correlated with the rheological data it can be observed the following: when the polymer fraction PEEK is constant in the NMP solution, and the weight fraction of PI-1 increased, the viscosity of the solution increase because the attachment level of the polymers chain was increased, leading at electrospun PEEK/PI-1 fibers with a high average fibers’ diameter.

### 3.4. Water Vapor Sorption Behavior

According to the IUPAC classification, the sorption/desorption isotherms for the electrospun PEEK/PI-1 fibers can be associated with the type IV curve describing sorption on mesoporous materials in which capillary condensation occurs [47,48,49]. The values of water vapor sorption capacities for all electrospun PEEK/PI-1 fibers are presented in Figure 7 and Table 3. The isotherms present a small type of H1 hysteresis between sorption and desorption branches [50,51]. If we look at Table 3, we can observe that the values of water vapor sorption capacity are higher for the electrospun PEEK/PI-1 (0.05–0.25) compared with the electrospun PEEK/PI-1 (0). This behavior correlates with the increase in concentration vs. activity of the sorption isotherm (Figure 7) and the presence of non-Fickian relaxations, indicating the occurrence of morphological changes in the polymer matrix as a result of swelling-induced plasticization. The electrospun PEEK/PI-1 (0.15) fibers exhibited the higher water sorption and desorption capacity. Thus, the electrospun PEEK/PI-1 (0.05, 0.1 and 0.25) fibers had close water sorption and desorption uptakes.

### 3.5. Thermal Characterization of the Electrospun PEEK/PI-1 Fibers

In order to investigate the *T_g_*^’^s of the samples, differential scanning calorimetry measurements were performed. The DSC curves for the electrospun PEEK/PI-1 fibers are presented in Figure 8. The *T_g_* values of the PEEK/PI-1 (0) and PI-1 were equal at 221.49 °C and 203.34 °C, respectively. Meanwhile, the value for the *T_g_* of the electrospun PEEK/PI-1 fibers varied between these two *T_g_* values. A slight decrease in *T_g_* values could be observed in the case of the electrospun PEEK/PI-1 fibers compared with electrospun PEEK/PI-1 (0) fibers. Probably, the introduction of the PI-1 decreased the polymer chain interaction, thus decreasing the *T_g_* values. Actually, the presence of the PI-1 in the electrospun PEEK fibers played the role of plasticizers, leading to the low temperatures of the glass transition. The presence of a single *T_g_* in the electrospun samples is ascribed to the good interfacial interaction between PEEK and PI-1.

The thermal stability of the electrospun PEEK/PI-1 fibers was investigated by thermogravimetric analysis (TGA). In Table 4 are listed the most important parameters obtained from the TG (Figure 9a) and DTG (Figure 9b) curves. From Figure 9a, it can be observed that with the introduction of PI-1 into PEEK, the initial thermal decomposition temperature decreased. The *T_5%_* value of the PEEK/PI-1 (0) was 423 °C, which is 222 °C higher than that of the PI-1. Meanwhile, the *T_5%_* values of the electrospun PEEK/PI-1 (0.1–0.25) fibers are in the range of these two limitations (325–414 °C). If *T_30%_* is taken as the thermal stability criterion, the series of the electrospun fibers for the thermal stability increase is as follows:PEEK/PI-1 (0.25) < PEEK < PEEK/PI-1 (0.2) ≈ PEEK/PI-1 (0.1) < PEEK/PI-1 (0.15) < PEEK/PI-1 (0.05)

The results exhibited that the introduction of 41.6 wt% PI-2 in the PEEK leads to a decrease in thermal stability. The effects of the PI-1 on thermal decomposition behavior are closely related to the wt% of the PI-1 and the degree of miscibility between these 2 components (PEEK and PI-1). In the nitrogen atmosphere, the electrospun PEEK/PI-1 fibers exhibit four weight loss stages in the intervals 50–70 °C, 100–200 °C, 350–500 °C and 550–750 °C. The first stage that occurs at temperatures below 100 °C (first marked area in Figure 9b) can be attributed to the evaporation of preabsorbed water and solvent from the electrospun PEEK/PI-1 fibers [52]. The second stage of the decomposition corresponded to the release of small molecules and the breakdown of the weak bonds such as P–O–C and P–C in the case of PEEK/PI-1 composite nanofiber membranes [19,20,53,54]. The third stage of thermal degradation, which correlates with the main degradation peak (Figure 9b), was situated in the interval 350–500 °C and it was due to the decomposition of ether units and cycloaliphatic moieties (phenolphthalein-based), which were more sensitive to degradation. The main finding here is that the introduction of smaller quantities of PI-1, in samples PEEK/PI-1 (0.05) and PEEK/PI-1 (0.10), resulted in composites with reinforced thermal stability, as the value of the third peak of thermal decomposition is superior for the sample PEEK/PI-1 (0.05) (512 °C) when compared with the neat PEEK sample, which is 504 °C (Table 4, Figure 9b). Moreover, in the case of sample PEEK/PI-1 (0.10) with an added amount of PI-1, the value of temperature related to this thermal degradation step is substantially pronounced (489 °C) when compared with the neat PI-1 polyimide (413 °C) (Table 4, Figure 9b). The fourth stage can be attributed to the decomposition of the stronger bonds, C–C bonds and/or ether linkages and/or amines, leading to the slow degradation of the char [55].

The char yield at 900 °C was 33 wt% for the PEEK/PI-1 (0) (area highlighted in Figure 9a). The introspection of TG curves revealed that the introduction of PI-1 in PEEK enhanced the thermal stability at elevated temperatures, the best composites in the series were PEEK/PI-1 (0.05), PEEK/PI-1 (0.10) and PEEK/PI-1 (0.15), which had values of the resultant carbonaceous residue of approximately 62, 63 and 64 wt% at 900 °C, respectively. Meanwhile, as presented in Figure 9b, the introduction of PI-1 decreased the maximum mass loss rate. Compared with PEEK, PI-1 decomposed at lower temperatures. In the case of the electrospun PEEK/PI-1 fibers, the release of the phosphoric acid probably catalyzed the decomposition of PEEK and led to relatively stable intermediate products that resulted in a decrease in the *T_max_* at high temperatures. 

Using the data obtained from the TGA analysis, the heat resistance index *T_HRI_*, which indicates the ability of polymers to resist a heat flow, can be calculated by using the following equation [56,57,58]:*T_HRI_* = 0.49 [*T_5%_* + 0.6(*T_30%_* − *T_5%_*)]

The heat resistance index of the electrospun PEEK/PI-1 fibers is shown in Table 4. The highest values of *T_HRI_* for the electrospun PEEK/PI-1 fibers were obtained for the samples PEEK/PI-1 (0.05). Moreover, can be observed from Table 4 that the *T_HRI_* increased with the increase of the PI-1 ratio into PEEK, but, in the case of the electrospun PEEK/PI-1 (0.25) can be observed a reduction of these parameters.

### 3.6. Water Absorption

The water uptake in the electrospun PEEK/PI-1 fibers was calculated by the weight difference before and after the small pieces of membrane were immersed in distilled water for 48 h at room temperature by using the following equation: $$WU\;\left(\%\right)=\left(\frac{W_{wet}-W_{dry}}{W_{dry}}\right)*100$$
where WU water uptake, $W_{wet}$ and $W_{dry\;}$ are the masses of the electrospun PEEK/PI-1 fibers (g) before and after water absorption, respectively. There are the following three mechanisms governing the water diffusion in polymeric composites: (i) the diffusion of water molecules into the vacant micro-spaces between polymer chains; (ii) capillary transport into the interfacial voids between the fiber and the polymer matrix; (iii) diffusion into the micro-fissured area of the polymer matrix [59]. Table 3 presents the water uptake values of the electrospun PEEK/PI-1 fibers. The electrospun PEEK/PI-1 (0) fibers have a water uptake equal with 1021.66 (%). The water uptake decreased for the electrospun PEEK/PI-1 (0.05–0.25) as the PI-1 content increased because it led to the decreased interfacial voids between composite fibers. For the electrospun PEEK/PI-1 (0.25) fibers, the water uptake significantly increased to 1297.61%, not only because the interfacial voids could absorb water but also because the hydrophilic PI-1 structure could accommodate an additional amount of water. 

### 3.7. Mechanical Properties

Figure 10 shows the stress-strain curves and the main mechanical properties of electrospun fibers with and without added PI-1. The presence of PI-1 in the electrospun PEEK/PI fibers led to improved mechanical properties (tensile strength, elastic modulus, elongation and ultimate tensile force). Thus, it was observed that a higher content of PI-1, up to 30%, induced an increase in elongation at break (Figure 10b) and the ultimate tensile force (Figure 10c), probably due to a slight plasticizing effect of PI-1 as a result of a good interaction between PI-1 and PEEK by hydrogen bonds. This effect also led to an increase in the elastic modulus from 27.5 MPa (PEEK/PI-1 (0)) to 44.02 MPa (PEEK/PI-1 (0.15)) (Figure 10d). The increase in PI-1 amount in the composition of the fibers to 41.6% PI-1 was observed in the stiffening of the PEEK/PI-1 fibers, with the consequence of a decrease in elongation at break and a slight increase in the modulus of elasticity. This aspect could be attributed both to the increase in electrostatic interactions by hydrogen bonds between the polymer chains, and the slight increase of the fibers’ diameter from 211–252 nm to 304 nm (Table 1). According to the mechanical behavior, PEEK/PI-1 with 30% PI-1 could be considered the ideal mixture for obtaining electrospun fibers with application in firefighting protective clothing.

### 3.8. Thermal Treatments of the Electrospun PEEK/PI-1 Fibers

When the temperature exceeds 260 °C, most materials used in various fields of application combust in the presence of oxygen, resulting in material property loss. In this respect, the electrospun PEEK/PI-1 fibers were exposed to 260 °C for 20 min, and then the structural morphology was investigated. The annealing temperature was above the *T_g_* of the electrospun PEEK/PI-1 (0) fibers, which was equal to 221.49 °C. Comparing the SEM images of the electrospun PEEK/PI-1 fibers (Figure 6) with the SEM images of the electrospun PEEK/PI-1 fibers exposed at 260 °C for 20 min (Figure 11), it can be observed that the surface morphology is not affected by the thermal annealing treatment. So, these new materials can be used for a long time in harsh conditions without losing their structural integrity.

### 3.9. Heat Transmission Measurements on Electrospun PEEK/PI-1 Fibers

To further investigate the thermal properties of electrospun PEEK/PI-1 fiber membranes, heat flux measurements were conducted according to the standard EN ISO9151:2016 [40]. The thermal energy transferred through the fabric specimen was processed using a copper sensor placed on the top of the fiber membrane. Figure 12 exhibits the temperature-time dependence curves. The near-linear shape of the curves presented obvious similarity, yet the recorded temperatures slightly decreased in the series as the content of DOPO-containing polyimide increased. In the case of some of the tested samples, the recorded outputs showed short nonlinear temperature-time regions, which may be correlated with the consumption of the material in the presence of a flame, in some zones, as was observed by visual introspection of the fabrics after running the tests (microphotographs insets in Figure 12). The heat transfer index ranks the ability of clothing assemblies to delay heat transfer from a flame. It is commonly used for textiles used in personal protective equipment.

The heat transfer index, which was calculated as the mean of the times required for a temperature rise of (24 ± 0.2) °C rounded up to the nearest integer number, varied in the series in the range of 4.51–5.1. If the thickness of the tested samples is considered (the measured value was approximately 0.06 mm in the series), the obtained values may be considered very attractive, as values of 14–17 have been reported in the literature for multilayered fabric assemblies [60]. Moreover, other authors have categorized the performance level of fabric specimens that are tested according to the ISO 9151:2016 (2016) standard in: Level 1: 4 s ≤ HTI < 10 s, Level 2: 10 s ≤ HTI < 20 s, or Level 3: HTI ≥ 20 s [61]. Thus, the best candidates in the series appealing for protective clothing applications, according to the heat transmission test, are the samples denoted as PEEK/PI-1 (0.15) and PEEK/PI-1 (0.20). Thus, the visual observation of the after tests samples revealed uniform and integer appearance on the whole surface exposed to the direct flame, without carbonized zones or even with darkened colors.

## 4. Conclusions

In the current study, a poly(ether-ether-ketone) containing bulky phenolphthalein groups was mixed with certain amounts of a polyimide containing two phosphaphenanthrene units per structural units. The electrospinning process was applied with the purpose of developing innovative nanofiber membranes with high-performance properties appealing to specific applications where high thermal stability and fire-resistant materials are needed. Before performing the electrospinning procedure, the solutions of the mixed polymers were subjected to rheology measurements in order to establish the optimal conditions that allowed the formation of smooth and beaded free nanofibers. The structural confirmation, morphology and thermal properties of the as-prepared nanofiber membranes have been performed by means of FTIR, SEM, TG, DSC and heat transmission measurements. The average fibers’ diameters of the electrospun PEEK/PI-1 fibers increased as the fraction of PI-1 in the precursor solution increased. The heat flux measurements conducted according to the standard EN ISO9151:2016 revealed slightly decreased temperatures in the series as the content of DOPO-containing polyimide increased. The superior mechanical performances in the series were evidenced for the samples containing 30 wt% polyimide. Given the higher char yield at 900 °C values obtained in the mixed PEEK/PI-1 series (51–64%) when compared to the sample containing only PEEK (33%), and the non-burning behavior of the membranes containing 30 and 36.4 wt% polyimide, respectively, we may consider these materials as candidates for further development of multilayer assemblies designed for advanced fire-fighting protective clothing applications.

## Figures and Tables

**Figure 1 polymers-14-05501-f001:**
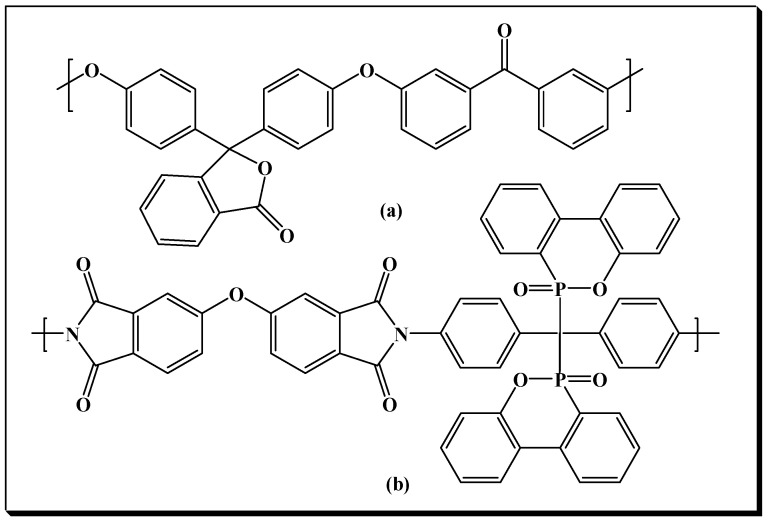
Structure of phenolphthalein-based PEEK (**a**) and phosphorus-containing polyimide PI-1 (**b**).

**Figure 2 polymers-14-05501-f002:**
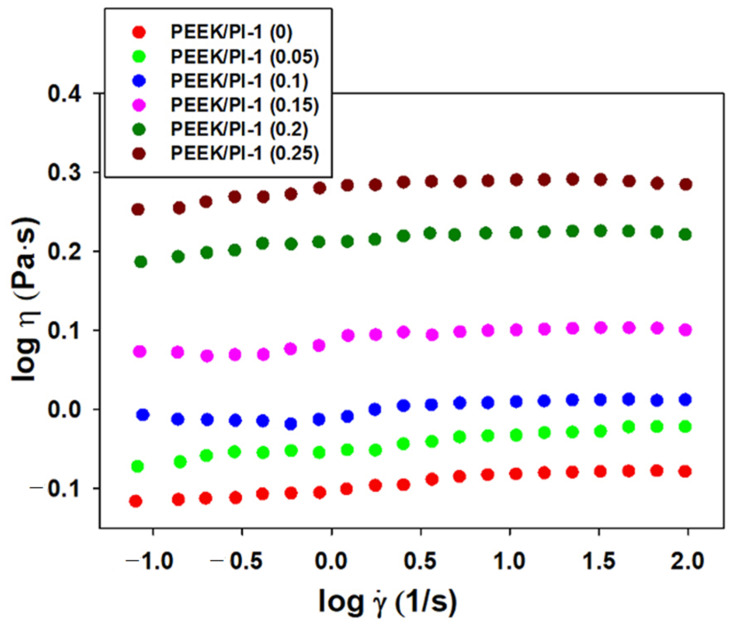
Rheological profile illustrated by double-logarithmic plots of dynamic viscosity versus shear rate for PEEK and PEEK/PI-1 studied systems at various mixing ratios and 25 °C.

**Figure 3 polymers-14-05501-f003:**
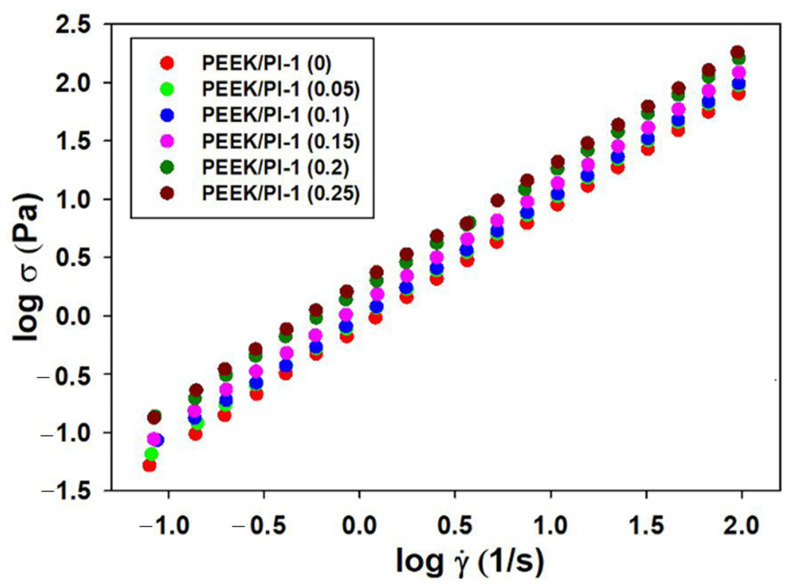
Ostwald–De Waele model represented by double-logarithmic plots of shear stress versus shear rate for PEEK and PEEK/PI-1 studied systems at various mixing ratios and 25 °C.

**Figure 4 polymers-14-05501-f004:**
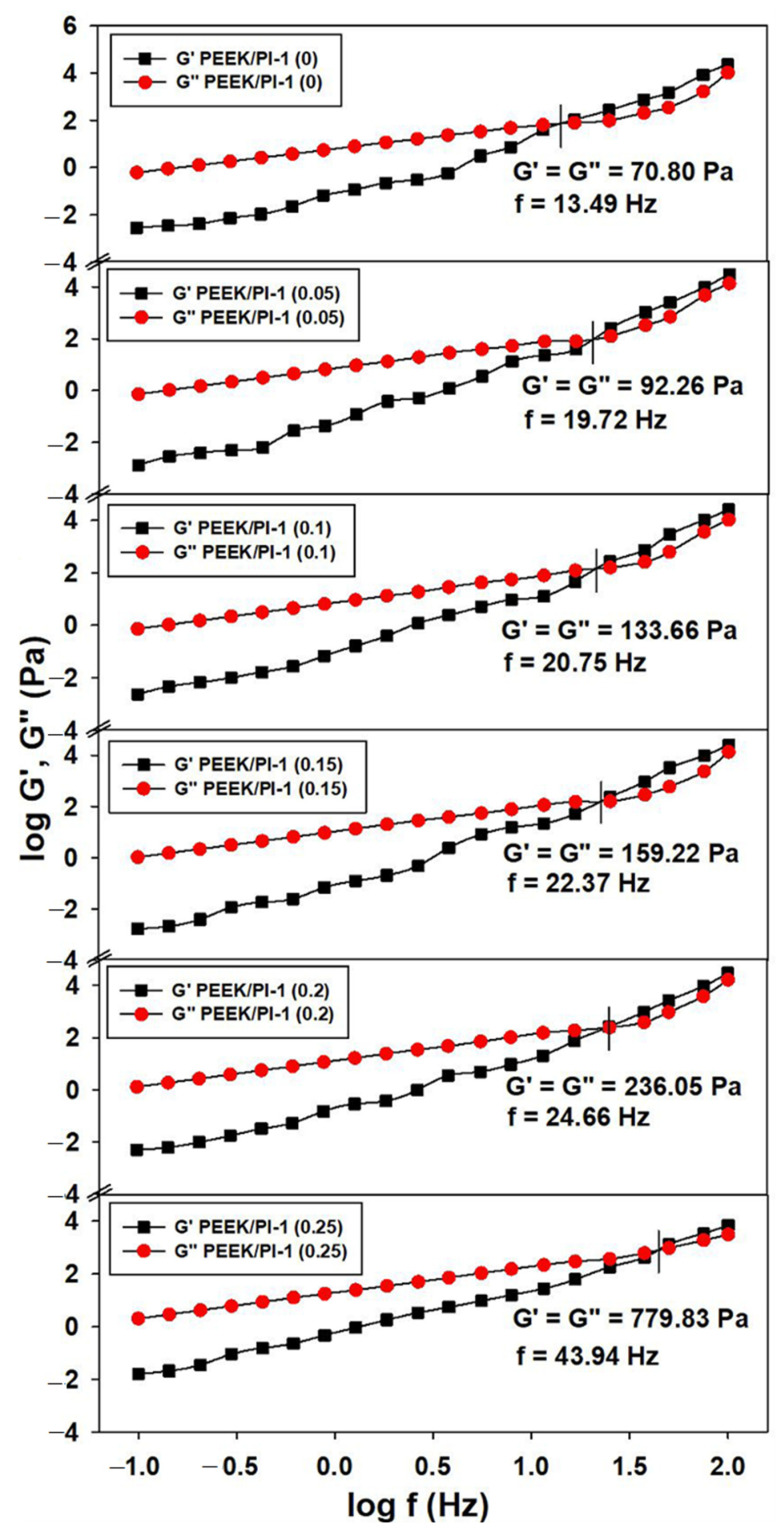
Double-logarithmic plots of shear moduli (G′ and G″) versus oscillatory frequency (f) for PEEK and PEEK/PI-1 studied systems at various mixing ratios and 25 °C.

**Figure 5 polymers-14-05501-f005:**
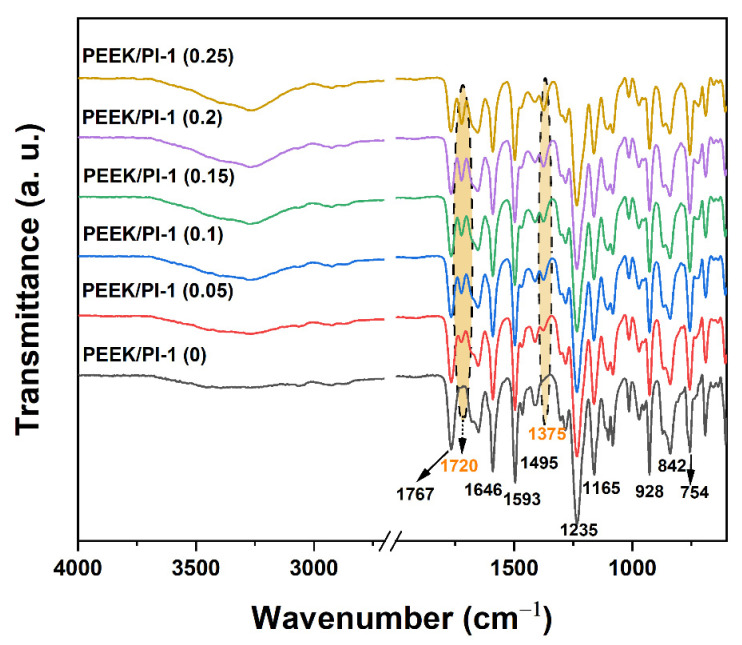
FTIR spectra of the studied samples.

**Figure 6 polymers-14-05501-f006:**
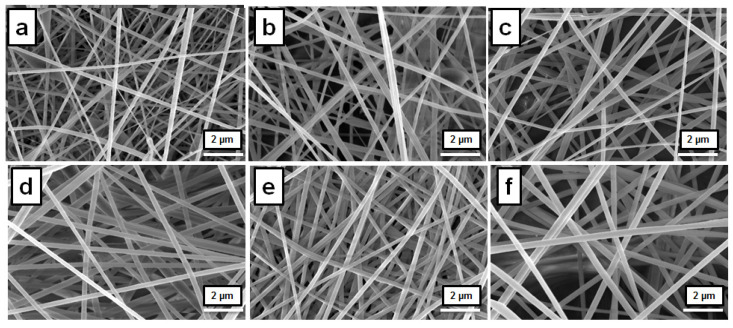
SEM images for the PEEK/PI-1 (0) (**a**), PEEK/PI-1 (0.05) (**b**), PEEK/PI-1 (0.10) (**c**), PEEK/PI-1 (0.15) (**d**), PEEK/PI-1 (0.20) (**e**), PEEK/PI-1 (0.25) (**f**).

**Figure 7 polymers-14-05501-f007:**
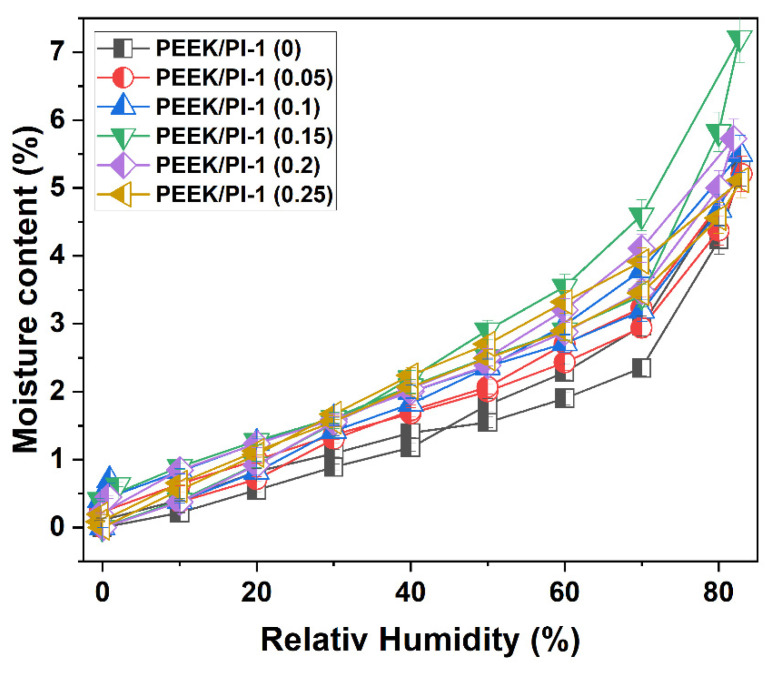
The shapes of the moisture sorption/desorption isotherms for the electrospun PEEK/PI-1 fibers.

**Figure 8 polymers-14-05501-f008:**
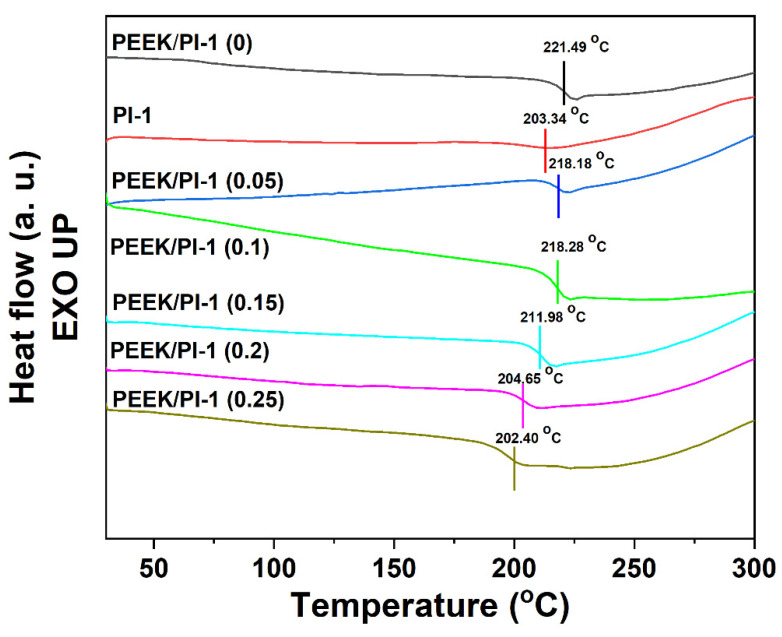
DSC curves for the electrospun PEEK/PI-1 fibers.

**Figure 9 polymers-14-05501-f009:**
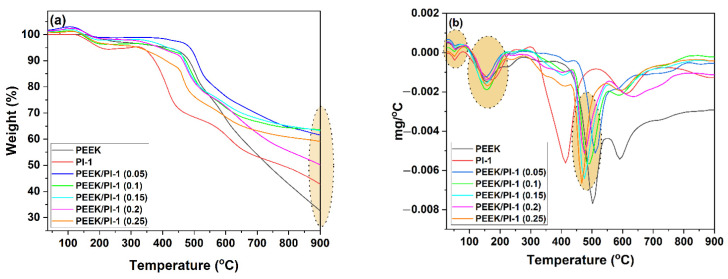
TG (**a**) and DTG (**b**) curves of the electrospun PEEK/PI-1 fibers.

**Figure 10 polymers-14-05501-f010:**
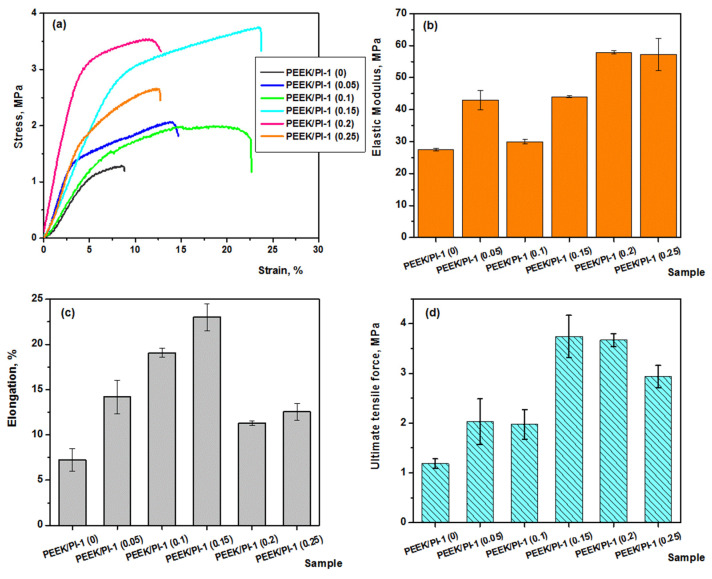
Mechanical properties for the electrospun PEEK/PI-1 fibers: stress–strain curves (**a**); elastic modulus (**b**); elongation (**c**) and ultimate tensile test (**d**).

**Figure 11 polymers-14-05501-f011:**
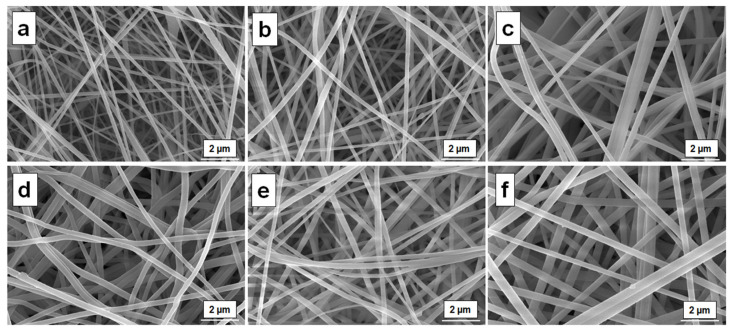
SEM images for the PEEK/PI-1 (0) (**a**), PEEK/PI-1 (0.05) (**b**), PEEK/PI-1 (0.1) (**c**), PEEK/PI-1 (0.15) (**d**), PEEK/PI-1 (0.2) (**e**), PEEK/PI-1 (0.25) (**f**) heated at 260 °C for 20 min.

**Figure 12 polymers-14-05501-f012:**
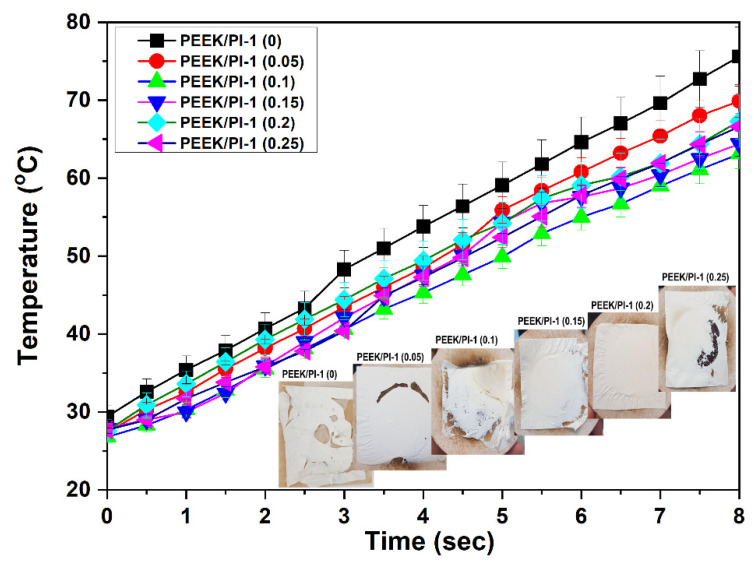
Heat transmission properties for the electrospun PEEK/PI-1 fibers.

**Table 1 polymers-14-05501-t001:** Flow behavior index, n, the consistency index, K ($Pa\cdot s^{n}$) and the corresponding regression coefficients (r^2^), for PEEK/PI-1 casting solutions in NMP at different mixing ratio.

Sample	n	K	r^2^
PEEK/PI-1 (0)	1.027	0.776	0.999
PEEK/PI-1 (0.05)	1.024	0.916	0.999
PEEK/PI-1 (0.10)	1.015	0.975	0.999
PEEK/PI-1 (0.15)	1.021	1.189	0.999
PEEK/PI-1 (0.20)	1.013	1.611	0.999
PEEK/PI-1 (0.25)	1.017	1.819	0.999

**Table 2 polymers-14-05501-t002:** Electrospinning condition and the average fiber diameters.

Sample Codes	PEEK (mL)	Weight PI-1 (g)/(%)	Electrospinning Condition	Average Fiber Diameters (nm)
			15 wt% PEEK,	
PEEK/PI-1 (0)	2	0/-	22 kV ^a^, 20 cm ^b^, 20 µL/min ^c^, ±25 °C ^d^, ±49% ^e^	211 ± 48
			15 wt% PEEK, 0.05 g PI-1,	
PEEK/PI-1 (0.05)	2	0.05/12.5	22.7 kV, 20 cm, 20 µL/min, ±21 °C, ±56%	212 ± 32
			15 wt% PEEK, 0.1 g PI-1,	
PEEK/PI-1 (0.10)	2	0.1/22.2	22.9 kV, 20 cm, 20 µL/min, ±23 °C, ±60%	215 ± 53
			15 wt% PEEK, 0.15 g PI-1,	
PEEK/PI-1 (0.15)	2	0.15/30	22.2 kV, 20 cm, 20 µL/min, ±21 °C, ±45%	252 ± 52
			15 wt% PEEK, 0.2 g PI-1,	
PEEK/PI-1 (0.20)	2	0.2/36.4	22.2 kV, 20 cm, 20 µL/min, ±21 °C, ±45%	237 ± 54
			15 wt% PEEK, 0.25 g PI-1,	
PEEK/PI-1 (0.25)	2	0.25/41.6	22.2 kV, 20 cm, 20 µL/min, ±20 °C, ±47%	304 ± 45

^a^ voltage; ^b^ the distance between the tip of the needle and the collector; ^c^ the flow rate of the solution; ^d^ temperature; ^e^ relative humidity (RH).

**Table 3 polymers-14-05501-t003:** Surface parameters evaluated based on adsorption/desorption isotherms: moisture sorption capacity; W, final weight, water uptake (WU) and BET data.

Samples	Weight (%)	BET		WU (%)
		Area (m^2^/g)	Monolayer (g/g)	
PEEK/PI-1 (0)	5.2891	55.242	0.0157	1021.66
PEEK/PI-1 (0.05)	5.2077	64.021	0.0182	876.92
PEEK/PI-1 (0.1)	5.5010	70.561	0.0201	809.52
PEEK/PI-1 (0.15)	7.2118	102.870	0.0293	702.70
PEEK/PI-1 (0.2)	5.7263	94.188	0.0268	570.49
PEEK/PI-1 (0.25)	5.1129	72.913	0.0207	1297.61

**Table 4 polymers-14-05501-t004:** TG and DTG data of the electrospun PEEK/PI-1 fibers.

Samples	*T_5%_*^a^ (°C)	*T_30%_* ^b^ (°C)	*T_HRI_*^c^ (°C)	*T_max_* (°C)	Residue at 900 °C
PEEK/PI-1 (0)	423	591	257	60 ^d^; 156 ^e^; 504 ^f^; 591 ^g^	33
PEEK/PI-1 (0.05)	484	695	299	50 ^d^; 151 ^e^; 512 ^f^; 689 ^g^	62
PEEK/PI-1 (0.10)	396	627	262	50 ^d^; 156 ^e^; 489 ^f^; 589 ^g^	63
PEEK/PI-1 (0.15)	414	653	273	50 ^d^; 152 ^e^; 474 ^f^; 595 ^g^	64
PEEK/PI-1 (0.20)	396	625	261	60 ^d^; 159 ^e^; 475 ^f^; 630 ^g^	51
PEEK/PI-1 (0.25)	325	583	235	60 ^d^; 159 ^e^; 467 ^f^; 594 ^g^	59

^a^ temperature where 5 wt% of the weight was lost; ^b^ decomposition temperature of 30% weight loss; ^c^ heat resistance index; ^d^ the first DTG peak; ^e^ the second DTG peak; ^f^ the three DTG peak; ^g^ the four DTG peak.

## Data Availability

Not applicable.

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
