# Peer review of "Electrospun Nanofibers Based on Polymer Blends with Tunable High-Performance Properties for Innovative Fire-Resistant Materials"

_polymers, 2022, doi:10.3390/polym14245501_

Round 1

Reviewer 1 Report

The manuscript reports the fabrication of micro-/nano-scaled non-woven fibrous membranes based on various ratios of a poly(ether-ether-ketone) (PEEK) and a phosphorus-containing polyimide as fire-resistant materials. The manuscript is well written and organized; however, revision is needed according to the following comments.

1)     The novelty of the work needs to be highlighted clearly in the introduction as compared to other articles published recently in the year 2020-2022.

2)     The characterization analysis of the synthesized materials is looking incomplete. Authors should provide the scientific explanation with other analytical characterization techniques regarding structural identification and surface morphology such as X-ray photoelectron spectroscopy (XPS) and atomic force microscopy (AFM).

3)     Authors should reports every peeks and their scientific interpretation in details in the spectra of FTIR, DSC curves, TG curves, DTG curves.

4)     Please provide comparative details (Tables) of the fire-resistant efficacy of the developed materials as compared to other reported in literature.

5)     Please provide the error bar in each figures which are representing experimental results.

6)     Quality of figures are very poor, should be improved in revised manuscript.

7)     English of the manuscript should be revised carefully.

Therefore, my recommendation is resubmission of the manuscript with a major revision in the suggested points, before being considered for publication in this Journal.

Author Response

Reply to the comments of Reviewer 1:

Thank you very much for reviewing our manuscript. We carefully revised the manuscript according to your valuable comments. Our point-by-point replies to the comments of the reviewer and the yellowed changed parts in the revised manuscript were specified below:

The manuscript reports the fabrication of micro-/nano-scaled non-woven fibrous membranes based on various ratios of a poly(ether-ether-ketone) (PEEK) and a phosphorus-containing polyimide as fire-resistant materials. The manuscript is well written and organized; however, revision is needed according to the following comments.

Q1.    The novelty of the work needs to be highlighted clearly in the introduction as compared to other articles published recently in the year 2020-2022.

A1. In the introduction section certain sentences were introduced to highlight the novelty of the current paper. For the first time in the literature PEEK was blended with a polyimide containing phosphorus. The presence of phosphorus atoms in the polyimide in our case is novel. We've never come across PEEK/P-containing polyimide nanofibers that are flame retardant. Furthermore, PEEK has good mechanical properties and thermal resistance, while the presence of nitrogen and phosphorus atoms in the polyimide increases the decomposition residue, improving the flame resistance of the nanofibers.

Q2. The characterization analysis of the synthesized materials is looking incomplete. Authors should provide the scientific explanation with other analytical characterization techniques regarding structural identification and surface morphology such as X-ray photoelectron spectroscopy (XPS) and atomic force microscopy (AFM).

A2. We were not able to perform AFM measurements for these samples. Instead, the samples were investigated by profilometry. We only managed to investigate 2 samples by profilometry, one of the reasons why we failed to do this study being the fact that the membranes become very electrified and stick to the cantilever. The cantilever used in AFM is much more sensitive/fragile, catching the electrospun fibers would definitely lead to the destruction of the cantilever. This is one of the reasons why we did not perform this analysis.

Below we present the data obtained from the profilometry studies for the samples where we managed to record. In the case of the electrospun sampes (PEEK/PI-1 (0), PEEK/PI-1 (0.05), PEEK/PI-1 (0.2) and PEEK/PI-1 (0.25)), they were pulled by the stylus itself.

PEEK/PI-1 (0.1)

Ra=125 nm

Rq=165 nm

PEEK/PI-1 (0.15)

Ra=114 nm

Rq=147 nm

In terms of X-ray photoelectron spectroscopy (XPS), such a study would have been interesting if our samples had different metals or, at the very least, different particles. But in our case, we don't expect to gain further knowledge about mixed PEEK with DOPO-containing polyimide. DSC and the rheological studies clearly show that the two systems chosen for this study are compatible. For the reviewer attention we present here the investigation of the electrospun PEEK/PI-1 fibers by EDX that is a qualitative analysis, to identify the elements in the sample - the chemical structure cannot be observed, nor is it a sensitive surface technique like XPS, which is based on the bond energy between atoms. EDX information is taken from the depth, from the volume and many artifacts can be reported during the analysis. This is the reason why we prefer not to include this analysis in the article.

Q3. Authors should reports every peeks and their scientific interpretation in details in the spectra of FTIR, DSC curves, TG curves, DTG curves.

A3. We have checked the referred paragraphs in the Results and discussion Section and we did our best to improve the scientific interpretation of the reference peaks.

Q4. Please provide comparative details (Tables) of the fire-resistant efficacy of the developed materials as compared to other reported in literature.

A4. At the moment it is difficult to find data in the literature to compare with them. In terms of fire-resistant efficacy of materials designed for firefighting we speak about multilayered fabric assemblies. In this respect, we may say that our study might considered a preliminary study with exceptional result, as the best candidates in the series literary do not burn when exposed to the flame for 8 seconds, according to the ISO9151 standard. It is expected, and we will give a try to develop further the nanofiber membranes presented in the current article. Thus, the construction of multilayered assemblies is our main interest, constructs which have to include as a fire-resistant layer one of the as developed PEEK/PI-1 nanofiber membranes presented in the current study.

Q5. Please provide the error bar in each figures which are representing experimental results.

A5.  The error bar was added in the figures.

Q6. Quality of figures are very poor, should be improved in revised manuscript.

A6. In the new version, the figures were uploaded separately.

Q7. English of the manuscript should be revised carefully.

A7. The English was carefully revised in the manuscript.

Reviewer 2 Report

This paper mainly describes the preparation of novel flame retardant nanofiber membranes by PEEK and phosphate-containing PI blend. Authors have novel idea sand completed experiments. But there are still some parts for improvement. Please refer to the comments below.

Page 2 line 96: Electrospinning can also be used to prepare ion exchange membranes for energy conversion applications, please refer this review to get more useful information. "Electrospun composite proton-exchange and anion-exchange membranes for fuel cells." Energies 14.20 (2021): 6709

Page 4 line 176: Can we know the density of 15wt% PEEK? This helps calculate the mass ratio of PEEK to PI and the concentration of the blend solution

Page 6 line 269: I wish the author could explain what PEEK/PI-2 is?

Page 11 line 377 The author mentioned increase the interaction between polymers molecules. But Page 12 line 424 the mentioned the introduction of the PI-1 decreased the polymer chains interaction. I want to know what’s the right explanation.

From my naked eye, nanofiber diameter in Figure 6a is significantly more uneven than Figure 6e, but the CV values are the same for both. Does this mean that the CV value is not accurate enough to describe uniformity of nanofiber mat? I suggest authors using DiameterJ in ImageJ to automatically measure fiber diameter distribution and get the plot.

I feel like there are so many curves in Figure 7 that it's hard for me to distinguish the details of each one.

For Table 3, I hope authors can explain why the average fiber diameter of PEEK/PI-1 (0.15) is higher than that of PEEK/PI-1 (0.10) and PEEK/PI-1 (0.20), but the area of PEEK/PI-1 (0.15) measured by BET is the highest one.

In Table 4, PEEK/PI-1 (0.15) once again shows abnormally high values of T5% and T30% comparing with PEEK/PI-1 (0.10) and PEEK/PI-1 (0.20). Can the author explain why?

Author Response

Reply to the comments of Reviewer 2:

Thank you very much for reviewing our manuscript. We carefully revised the manuscript according to your valuable comments. Our point-by-point replies to the comments of the reviewer and the yellowed changed parts in the revised manuscript were specified below:

This paper mainly describes the preparation of novel flame retardant nanofiber membranes by PEEK and phosphate-containing PI blend. Authors have novel idea sand completed experiments. But there are still some parts for improvement. Please refer to the comments below.

Q1. Page 2 line 96: Electrospinning can also be used to prepare ion exchange membranes for energy conversion applications, please refer this review to get more useful information. "Electrospun composite proton-exchange and anion-exchange membranes for fuel cells." Energies 14.20 (2021): 6709

A1.  The reference has been cited in the introduction section.

Q2. Page 4 line 176: Can we know the density of 15wt% PEEK? This helps calculate the mass ratio of PEEK to PI and the concentration of the blend solution

A2. The density of the 15wt.%PEEK was investigated by a Density Kit achieved from the Mettler Toledo and was equal with 1.29 g/cm3.

Q3. Page 6 line 269: I wish the author could explain what PEEK/PI-2 is?

A3. Sorry, this was a mistake. The PEEK/PI-2 was corrected with PEEK/PI-1.

Q4. Page 11 line 377 The author mentioned increase the interaction between polymers molecules. But Page 12 line 424 the mentioned the introduction of the PI-1 decreased the polymer chains interaction. I want to know what’s the right explanation.

A4. The two referred paragraphs have been adjusted in the revised manuscript.

Q5. From my naked eye, nanofiber diameter in Figure 6a is significantly more uneven than Figure 6e, but the CV values are the same for both. Does this mean that the CV value is not accurate enough to describe uniformity of nanofiber mat? I suggest authors using DiameterJ in ImageJ to automatically measure fiber diameter distribution and get the plot.

A5. To avoid the confusion, we decide to erase the referred paragraph.

Q6. I feel like there are so many curves in Figure 7 that it's hard for me to distinguish the details of each one.

A6. Figure 7 has been modified.

Q7. For Table 3, I hope authors can explain why the average fiber diameter of PEEK/PI-1 (0.15) is higher than that of PEEK/PI-1 (0.10) and PEEK/PI-1 (0.20), but the area of PEEK/PI-1 (0.15) measured by BET is the highest one.

A7. From all the studies carried out, it could be observed that the best results were obtained for the electrospun PEEK/PI-1 (0.15) fibers. Probably, this is the maximum amount of PI-1 that can be added to PEEK, at which the compatibility between the two polymers is optimal. Next, we will choose this matrix to study in detail the comfort properties (such as: air permeability, water vapor transition rate, etc.) of the material, thus directing it towards practical applications. At higher concentration of PI-1 added to the PEEK, the viscosity of the solution was changed and also the average fibers diameters were affected.

Q8. In Table 4, PEEK/PI-1 (0.15) once again shows abnormally high values of T5% and T30% comparing with PEEK/PI-1 (0.10) and PEEK/PI-1 (0.20). Can the author explain why?

A8. As presented previously, at question 7, Probably, this is the maximum amount of PI-1 that can be added to PEEK, at which the compatibility between the two polymers is optimal.

Reviewer 3 Report

1. The NMR spectra of the polymers should be presented and peaks assigned and integrated.

2. Moisture, water uptake and studies of PEEK and PI type materials should be discussed and results compared.

3. The selection of the two polymers for the blend should be justified. There are many different polyimide and PEEK that could have been synthesized and used for this purpose.

4. The molecular weight of the synthesized polymers should be determined and GPC results reported. It is one of the most important characteristics of polymers that will determine the processability and durability of the final materials developed.

5. The optimization of the electrospinning should be shown in more detail in the manuscript. Figures 6, 11 presents already optimized fibers only. What are the limitations of the collector distance, voltage, solution viscosity, solvents etc?

6. Polyimide electropinning has been emergin recently which should be mentioned as prior art and demonstrate its importance on its own (10.1021/acsanm.1c03280; 10.1016/j.cej.2022.137821).

7. The use of PEEK in various areas is speading which is important to mentioned with some examples, it has a good potential (10.1016/j.memsci.2021.120015; 10.1039/D1TA03690D).

8. The authors should report benchmarks for the entire study, including the BET, moisture sorption, mechanical analysis, morphology, such as pure PEEK and pure PI materials, their characterization and performance.

Author Response

Reply to the comments of Reviewer 3:

Thank you very much for reviewing our manuscript. We carefully revised the manuscript according to your valuable comments. Our point-by-point replies to the comments of the reviewer and the yellowed changed parts in the revised manuscript were specified below:

Q1. The NMR spectra of the polymers should be presented and peaks assigned and integrated.

A1. The NMR spectra for the PI-1 was presented previously in the paper cited as the references in this paper. The 1H NMR data for the PEEK was added in the manuscript. We prefer not to present the spectra also in the manuscript, as the number of figures will increase.

NMR spectrum for PI-1, in the aromatic region.

NMR spectrum for PEEK, in the aromatic region.

Q2. Moisture, water uptake and studies of PEEK and PI type materials should be discussed and results compared.

A2. Water uptake measurements were conducted and introduced in the revised manuscript.

Q3. The selection of the two polymers for the blend should be justified. There are many different polyimide and PEEK that could have been synthesized and used for this purpose.

A3. We agree that the structural versatility of PEEKs and Polyimides is limitless. Nevertheless, the phenolphthalein-containing PEEK was chosen because its exceptional electrospinning behavior, while the polyimide is containing two side chained bulky phosphaphenanthrene group which bring on the table plenty of aromatic structures accompanied by the presence of phosphorus atom in the molecule, atom that is well known for improving flame resistance of certain materials with advanced flammability. The combination of the two polymers was expected to give nanofiber membranes fine and well deposited by electrospinning.

Q4. The molecular weight of the synthesized polymers should be determined and GPC results reported. It is one of the most important characteristics of polymers that will determine the processability and durability of the final materials developed.

A4. The requested data were introduced in the revised manuscript.

Q5. The optimization of the electrospinning should be shown in more detail in the manuscript. Figures 6, 11 presents already optimized fibers only. What are the limitations of the collector distance, voltage, solution viscosity, solvents etc?

A5. In fact, the parameters of the electrospinning process were investigated and optimized in advance and we chose the optimum electrospinning parameters in order to obtain the membranes with best characteristic. We consider that we reported sufficient data in the manuscript and we do not consider that the optimization process should be part of the current manuscript.

Q6. Polyimide electropinning has been emergin recently which should be mentioned as prior art and demonstrate its importance on its own (10.1021/acsanm.1c03280; 10.1016/j.cej.2022.137821).

A6. The Introduction Section has been improved citing the referred articles.

Q7. The use of PEEK in various areas is speading which is important to mentioned with some examples, it has a good potential (10.1016/j.memsci.2021.120015; 10.1039/D1TA03690D).

A7. The Introduction Section has been improved citing the referred articles.

Q8. The authors should report benchmarks for the entire study, including the BET, moisture sorption, mechanical analysis, morphology, such as pure PEEK and pure PI materials, their characterization and performance.

A8. For the pure PEEK this information is presented in the paper. Pure PEEK in qur case in the samples denoted PEEK/PI-1 (0). For PI-1 we do not present this information because this polyimide does not give fibers and moreover it was not possible to obtain a membrane by using only this polymer. Only if I do these analyzes on the polymer powder and, honestly, I don't know how I could compare them afterwards.

Round 2

Reviewer 1 Report

Accepted

Reviewer 2 Report

Authors have modified paper and answered the questions as suggested. My advice is accept.

Reviewer 3 Report

The points were addressed and article is publishable now.